# Digestibility, Blood Parameters, Rumen Fermentation, Hematology, and Nitrogen Balance of Goats after Receiving Supplemental Coffee Cherry Pulp as a Source of Phytochemical Nutrients

**DOI:** 10.3390/vetsci9100532

**Published:** 2022-09-28

**Authors:** Yudithia Maxiselly, Rawee Chiarawipa, Krit Somnuk, Puwadon Hamchara, Anusorn Cherdthong, Chanon Suntara, Rittikeard Prachumchai, Pin Chanjula

**Affiliations:** 1Department of Agronomy, Faculty of Agriculture, Universitas Padjadjaran, Sumedang 45363, Indonesia; 2Agricultural Innovation and Management Division, Faculty of Natural Resources, Prince of Songkla University, Songkhla 90110, Thailand; 3Department of Mechanical Engineering, Faculty of Engineering, Prince of Songkla University, Songkhla 90110, Thailand; 4Program of Animal Science, Faculty of Science and Technology, Suratthani Rajabhat University, Suratthani 84100, Thailand; 5Tropical Feed Resource Research and Development Center (TROFREC), Department of Animal Science, Faculty of Agriculture, Khon Kaen University, Khon Kaen 40002, Thailand; 6Animal Production Innovation and Management Division, Faculty of Natural Resources, Hat Yai Campus, Prince of Songkla University, Songkhla 90110, Thailand

**Keywords:** coffee residue, phytonutrient substance, ruminal fermentation, greenhouse gas, goat

## Abstract

**Simple Summary:**

Coffee pulp is a typical byproduct in nations where coffee is cultivated, particularly Thailand. Such phytonutrients as tannin, saponin, and chlorogenic acid are abundant in this residue. Ruminants’ rumen fermentation and feed utilization may both be improved by these substances. Our study demonstrates that feeding goats coffee pulp improved animal digestibility and rumen fermentation activity without affecting feed intake or blood metabolite levels.

**Abstract:**

This research examines the impact of adding dried coffee cherry pulp (CoCP) to goat feed on the digestibility of the feed, rumen fermentation, hematological, and nitrogen balance. A goat feeding experiment employed four male crossbreds (Thai Native × Anglo Nubian) aged 12 months and weighing 21.0 ± 0.2 kg each. The treatment was conceived as a 4 × 4 Latin square with four specific CoCP levels at 0, 100, 200, and 300 g/day. Dry matter intake (DMI), organic matter intake (OMI), and crude protein intake (CPI) were unaffected by the addition of CoCP. However, across treatment groups, there was a linear increase in ether extract intake (EEI) (*p* < 0.01), neutral detergent fiber intake (NDFI) (*p* = 0.06), and acid detergent fiber intake (ADFI) (*p* = 0.04), as well as a quadratic effect on DMI% BW (*p* = 0.04). The findings showed that rumen temperature, pH, ammonia-nitrogen, or pack cell volume did not change with CoCP supplementation. Total volatile fatty acid showed linear effects on acetate (*p* = 0.03) and was quadratically affected by propionate concentration (*p* = 0.02), acetate to propionate ratio (*p* = 0.01), acetic plus butyric to propionic acid ratio (*p* = 0.01), and methane estimation (*p* = 0.01). With increased CoCP supplementation, there was a linear decrease in protozoa count by about 20.2% as the amount of CoCP supplemented increased (*p* = 0.06). CoCP supplementation in animal feed resulted in a linear decrease in urinary nitrogen (*p* = 0.02) and a quadratic effect on absorbed nitrogen (*p* = 0.08) among treatment groups, with greater N utilization values found in goats fed 200 g/d CoCP. In light of this, supplementing CoCP into animal feed may improve animal digestion and rumen fermentation effectiveness while having no effect on feed intake, rumen microbes, or blood metabolites.

## 1. Introduction

A major issue for the twenty-first century is ensuring a sustainable global food supply. The worldwide meat consumption will grow by 58% by 2050 [1]. Because it provides an adequate number of non-limiting amino acids and a balanced fatty acid profile, the flesh of small ruminants is one of the most significant sources of animal protein. Small ruminant production in tropical nations, especially Thailand, has been severely impacted by the degradation of grazing habitats and the growing expense of traditional concentrate diets [2]. For long-term sustainability and the development of shrewd livestock feeding practices, it is therefore imperative to give priority to research into alternative feed suppliers in low-input production systems such as those in Thailand [1]. The use of agro-industrial byproducts in livestock husbandry is essential in both emerging and developed nations. The agro-industrial application is gaining popularity because it offers phytonutrient sources (PTN), which are beneficial to animal nutrition and health [3].

Coffee pulp is a waste product that is plentiful in coffee-growing areas. The first by-product of coffee processing is coffee pulp (CoP) or coffee cherry pulp (CoCP), which is extracted from the grain or bean using either a wet or dry method. According to Gouvea et al. [4], CoP is a residue formed during coffee wet processing that accounts for roughly 40–50% of the wet weight of the fresh fruit cherry. [5]. Coffee trash production is estimated to be around 15 million tons per year [6]. CoCP has a high concentration of PTN and antioxidant compounds such as caffeine, saponin, phenolic substances, chlorogenic acid, and tannin content, which varies depending on the coffee species, stage of development, farming practices, and extraction process. [7]. According to certain researchers, PTN (such as tannin and saponin) is related to a decrease in protozoa and a decrease in methane (CH_4_) gas [3,7,8]. Protozoa are primary ruminal hydrogen (H_2_) providers, and the H_2_ produced is largely transformed to CH_4_ by methanogens found inside protozoa or on their surface [5]. Additionally, CoCP contains significant amounts of neutral detergent fiber (NDF, 41.5–46.0%), acid detergent fiber (ADF, 37.8–41.0%), crude protein (CP, 9.7–10.3%), 1.0–2.0% of lipids, and 45.0–89.0% of carbohydrates (including pectin), suggesting the possibility of using it as a ruminant feed [8,9,10]. The contents of P, K, Ca, and Mg in coffee pulp were found to be 2.48, 25.13, 4.10, and 1.39 g/kg, respectively, whereas Fe and Mn were found to be 77 and 46 mg/kg [11]. The CoCP can be used in the diets of ruminants, according to de Souza et al. [12], because tannins can also reduce the digestibility of carbohydrates and protein and prevent feed protein from rumen microbial degradation, enhancing animal production [13,14]. According to Barcelos et al. [15], supplementing coffee husk with 1.65 kg of DM/head/day did not influence weight gain in bulls. On dairy cows, some coffee by-products have been tested. Cow diets can contain up to 150 g/kg of coffee pulp without affecting output [16], or 25% of the diet at a 60:40 forage-to-concentration ratio [17,18]. As a result, CoCP has great potential as a ruminant feed ingredient.

The use of CoCP as a source of phytochemical content in goats, on the other hand, has gotten far less attention. We hypothesized that CoCP-containing PTN appears to affect rumen fermentation and positively affect performance in experimental animals. Therefore, the goal of this study was to see how adding CoCP as a phytochemical source affected goat rumen fermentation, nutritional digestibility, hematological, CH_4_ estimation, and nitrogen balance.

## 2. Materials and Methods

### 2.1. Preparing of Dried Coffee Cheery Pulp 

The Prince of Songkla University, Department of Plant Science, Faculty of Natural Resources, provided the Robusta CoCP used in this study. Machine extracts were used to remove the coffee beans from the CoCP after the ripe coffee production and harvesting were completed. CoCP was collected fresh and dried at 60 °C for 72 h before being ground in a Cyclotech Mill (Tecator, Hganäs, Sweden) to pass through a 0.2-mm mesh screen, packed in air-tight polyethylene bags, and stored in a freezing, dry place as recommended by Hagerman [19]. The quantities of tannins, phenolic compounds, flavonoids, saponin, caffeine, and antioxidant content in the CoCP were analyzed by Makkar et al. [20]. The total alkaloids were measured spectrophotometrically by Shamsa et al. [21]. The plant material was extracted using methanol (100 g). Following the extraction, a portion of the residue was melted in 2 N HCl and filtered again. 1 mL of this solution was transferred to a separatory funnel and rinsed with 10 mL of chloroform (3 times). After that, 5 mL of bromocresol green solution and 5 mL of phosphate buffer were added to the solution. After stirring the mixture, 1 to 4 mL chloroform was used to separate the resulting complex. In a 10-mL volumetric flask, the extracts were collected and diluted to volume using chloroform. The complex’s chloroform absorbance was measured at 450 nm (Unico Spectrophotometer, 2800 UV/VIS, NJ, USA).

### 2.2. Animals, Treatments, and Experimental Design

A goat feeding study was performed with four male crossbreds (Thai Native × Anglo Nubian) at 12 months of age (21.0 ± 0.2 kg). The treatment design was a 4 × 4 Latin square, with four different levels of CoCP. Based on a coffee literature review on byproduct utilization, daily CoCP supplementation at 0, 100, 200, and 300 g/day was used as the dietary treatment [12,15,16,17]. The diets were designed in accordance with NRC [22] standards in order to attain a daily gain of 100 g. The ruminants were given free access to feed (50 g/kg refusals) two times per day (07:00 a.m. and 4:00 p.m.). Before beginning the trial, the goats were injected (ivermectin) against parasites and given a vitamin injection with AD3E for prevention of vitamin deficiencies, stress and improvement of feed conversion in animals. Table 1 shows the components and chemical compositions of TMR (total mixed rations) and CoCP. To verify that the animals received all the supplements, CoCP was manually hand-mixed with 200 g of TMR feed before the morning meal, and more TMR was provided to individual animals later.

Individual pens (0.115 × 0.95 m) beneath well-ventilated sheds with a constant supply of water and mineral salts were used to raise the animals. The investigation was carried out over four 21-day intervals. The total animals were fed their respective diets *ad libitum* for the first 14 days before being moved to metabolism crates for total collection for the final 7 days, during which they were limited to 90 percent of their previous voluntary TMR feed intake and supplemented with varying amounts of CoCP to ensure overall feed consumption.

### 2.3. Sampling Method and Data Collection

#### 2.3.1. Fecal and Feed Sampling Methods

During the trial, the amount of feed supplied, the amount of CoCP, and orts samples were all recorded on a daily basis. To investigate nutritional digestibility, feed, orts, and fecal samples were collected from each individual goat using the total collection method during the last 7 days of each period. The obtained samples were dried at 72 °C, milled (1 mm screen using a Cyclotech Mill, Tecator, Sweden), and measured for dry matter (DM; ID 967.03), ash (ID 492.05), ether extract (EE; ID 455.08), and crude protein (CP; ID 984.13) by the protocols of the Association of Official Analytical Chemists [23]. Analyses of NDF and ADF were performed with α-amylase but without sodium sulphite according to Van Soest et al. [24] NDF and ADF were expressed inclusive of residual ash. The CoCP contained 94.5% DM, 11.6% CP, 0.97% ash, 3.52% EE, 43.6% NDF, 10.4% ADF of DM, 4.27 mg tannins/100 g DM, 2.61 mg saponins/100 g DM, 3.75 mg antioxidant activity content/100 g DM, 7.50 mg phenolic compound content/100 g DM, and 4.37 mg flavonoid content/100 g DM (Table 1). Using an automatic adiabatic bomb calorimeter, the feed gross energy (GE) content was calculated (AC 500, LECO Corporation, St. Joseph, Michigan, USA). The equation proposed by Robinson et al. [25] was used to compute metabolic energy (ME).

ME (MJ/kg DM) = 0.82 × [2.4 × CP + 3.9 EE × 1.8 × the rest of the OM] × in vitro organic matter digestibility (IVOMD).

#### 2.3.2. Urine Sampling Method

Whole urine was collected on the same days as feces in a plastic container treated with ten percent H_2_SO_4_. to keep the final pH below three and prevent nitrogen (N) extinction. Urine samples were taken at approximately 100 mL of total volume, frozen, and pooled at the end of each session using the AOAC total N measurement method [23].

#### 2.3.3. Rumen Fluid Sampling Method

On the final day of the data collection session, rumen fluid samples were taken at 0 and 4 h post-feeding. The oral stomach collection device consisted of a 90-cm polyvinyl chloride orogastric tubing with a 15-mL perforated plastic conical tube attached to one end that served as a rumen sieve. The other end of the stomach tube was attached to an electric vacuum pump. Each time, a stomach tube linked to a vacuum pump was utilized to extract approximately 50 mL of rumen fluid from the rumen center. The pH and temperature of rumen fluid were quickly measured using a portable pH and temperature meter (HANNA HI-8424 Portable pH/ORP Meter, Woonsocket, USA). Following that, rumen fluid samples were filtered through four levels of cheesecloth. A total of 45 mL of rumen fluid was collected and maintained in a plastic bottle with 5 mL of sulfuric acid solution to disrupt the microbial activity fermentation process (1 M). The mixture was centrifuged at 16,000× *g* for 15 min (Table Top Centrifuge PLC-02, FL, USA). Using a Kjeltech Auto 1030 Analyzer [23], the supernatant was tested for ammonia nitrogen (NH_3_-N), and volatile fatty acids (VFAs) were separated using a high-performance liquid chromatography method developed by Mathew et al. [26] (HPLC, Instruments by Controller, water model 600E, water model 484 UV detector, column Novapak C18, column size 4 mm 150 mm, mobile phase 10 mM H_2_PO_4_ (pH 2.5); ETL Testing Laboratory, Inc.). Estimation of ruminal CH_4_ using VFA proportions according to the equation of Moss et al. [27] is as follows: CH_4_ production = 0.45 (acetate) − 0.275 (propionate) + 0.4 (butyrate). Collect about 20 mL of ruminal fluid with a large-bore pipette (to determine for bacteria, protozoa, and zoospores) and add 1 mL to the formalin solution. This step yields a 1:10 dilution. After diluting, keep samples refrigerated. For protozoa counts, use a Sedgewick-Rafter chamber (S.I Scientific Supply Co., Ltd., Bangkok, Thailand), and add a cover slip. Add a grid to the eyepiece of the microscope. Then, to obtain the average per square, count 25 large squares (at random) and divide the number of protozoa counted by 25. It is possible that a higher dilution is required. Counting at 100 powers would most likely suffice. Use a Petroff-Hausser chamber for bacterial counts (Xinxiang Vic Science & Education Co., Ltd., Henan, China). To calculate the average number of bacteria per square, count one set of 25 squares, add the counts, and divide by 25. Use 600 power phase contrast [28]. Moreover, the total direct count of fungal zoospores was made using the methods of Galyean [28] based on the use of a haemacytometer (Booeco).

#### 2.3.4. Blood Metabolites

Blood samples (about 10 mL) were collected at 0 and 4 h post-feeding on the last day of the data collection period from the jugular vein into tubes containing 12 mg of EDTA, and the plasma was separated by centrifugation at 1500× *g* for 10 min (Table Top Centrifuge PLC-02, USA). The plasma was kept at 20 °C until it was analyzed for blood urea nitrogen (BUN), blood glucose (GLU), non-esterified fatty acids (NEFA), beta-hydroxybutyric acid (BHBA), and creatinine (Cr). These analyses were performed on a fully automatic biochemical analyzer using standard commercial kits supplied by Stanbio Laboratory (Taxas, USA); NEFA and BHBA were analyzed using kits from Randox (Crumlin, UK). The plasma concentrations of urea nitrogen and plasma glucose were measured by using a commercial kit (No. 640, Sigma Chemical Co., St. Louis, MO, USA). Packed cell volume (PCV) was determined using a microhematocrit method.

### 2.4. Statistical Analysis

All of the experimental data were analyzed to ANOVA for a 4 × 4 Latin square design using the general linear model (GLM) techniques [29]. The model was used to examine data.
Y_ijk_ = μ + M_i_ + A_j_ + P_k_ + ε_ijk_
where Y_ijk_ is the observation from animal j, receiving diet i, in period k; μ is the overall mean; M_i_ is the effect of the different levels of CoCP (i = 1, 2, 3, 4); A_j_ is the effect of animal (j = 1, 2, 3, 4); P_k_ is the effect of the period (k = 1, 2, 3, 4); and ε_ijk_ is the residual effect. To assess treatment effects on response variables, orthogonal polynomials (linear and quadratic) were used. p < 0.05 was used as the threshold for significance, with *p*-values of >0.05 < 0.10 indicating a trend approaching significance.

## 3. Results

### 3.1. Influence of CoCP on Feed Intake and Nutrient Digestibility

Feed intake, nutritional intake, and digestibility are shown in Table 2. Supplementing with CoCP showed no effect on DMI or dietary intake (OMI and CPI). However, a linear increase was found in EEI (*p* < 0.01), NDFI (*p* = 0.06), and ADFI (*p* = 0.04), and had a quadratic effect on DMI as % BW (*p* = 0.04), and DMI as BW^0.75^ (*p* = 0.06) among treatment groups. CoCP supplementation at 300 g/d decreased DM, OM, CP, NDF, and ADF digestibility while increasing EE digestibility (*p* < 0.05). There was no difference in estimated energy intakes (ME Mcal/DM/d) across treatments (*p* > 0.12). Nonetheless, energy intake (ME Mcal/kg DM) showed a quadratic influence (*p* = 0.01).

### 3.2. Influence of CoCP on Rumen Characteristics and Blood Metabolites 

Table 3 shows the rumen temperature, pH, ammonia nitrogen (NH_3_-N), PCV, and blood urea nitrogen (BUN) values from the rumen fermentation investigation. The addition of CoCP to the diet had no effect on rumen temperature, pH, NH_3_-N, or PCV. However, feeding CoCP at a rate of 200 g/d resulted in plasma GLU levels that were higher than those in the control group. BUN was lowered linearly (*p* = 0.01) after treatment with CoCP. NEFA (*p* = 0.01) and BHBA (*p* = 0.05) content also decreased linearly.

### 3.3. Influence of CoCP on Volatile Fatty Acid Profiles 

Table 4 and Figure 1 show the VFA from the rumen fermentation investigation. Total VFA had quadratic effects (*p* = 0.02), while acetate (*p* = 0.03), propionate concentration (*p* = 0.02), acetate to propionate ratio (*p* = 0.01), acetic plus butyric to propionic acid ratio (*p* = 0.01), and CH_4_ (*p* = 0.01) had linear effects. The addition of CoCP had no effect on the molar proportions of individual VFA, butyric acid, or other VFA (*p* > 0.05).

### 3.4. Influence of CoCP on Microbial Population

The amount of rumen microbes’ data is shown in Table 5. CoCP supplementation had no influence on ruminal bacterial populations or fungal zoospore counts (*p* > 0.05). Nonetheless, increased CoCP supplementation led to a linear reduction in protozoa (*p* = 0.06).

### 3.5. Influence of CoCP on N Balance

Figure 2 shows the N consumption, output, and use of N fed with various doses of CoCP. Total nitrogen intake and fecal excretion were not significantly different across treatments (*p* > 0.05). Nonetheless, CoCP supplementation had a linearly decreasing impact on urinary N (*p* = 0.02) and a quadratic impact on N absorption (*p* = 0.08) between treatment groups, with goats fed CoCP at 100–200 g/d having a higher value of N consumption.

## 4. Discussion

### 4.1. Nutrients and Phytochemical Composition of CoCP

Coffee waste has been the subject of much research, and it has ramifications for many parts of human life. Some studies have shown that coffee waste can be used as organic fertilizer and animal feed for rabbits, pigs, and fish [30]. When compared to coffee beans, the pericarp of coffee produces a high phenolic content, and it also has a similar tannin and polyphenol content to coffee seeds [31,32]. According to Table 1, CoCP possessed macronutrients such as crude protein, organic matter, and ether extract that were essentially identical to the composition of the basal diet, but it also had a lower ADF value than rice straw and the basal diet, indicating that this substance has greater nutrition as ruminant feed. Furthermore, phytochemical compounds or phytogenic feed additives (PFA) such as tannin, saponin, phenolic component flavonoids, and antioxidant activity at low concentrations were conceived by CoCP. Phytochemicals provide a wide range of benefits in animals, including lowering CH_4_, enhancing microbial activity, protein binding, regulating rumen diseases, and promoting fiber digestion [14,33,34]. Saponin is one of the phytochemical substances that may contribute to selective defaunation from the rumen microbial community, enhancing ruminant nitrogen utilization and perhaps improving growth and milk production [35]. The antioxidant effect also impacts the goat’s metabolism. Other studies using plants with high phenolic content found that processing plant secondary compounds increased the synthesis of biotransformation enzymes in goats [36]. The flavonoid component of the plant has been shown to influence rumen microbial metabolism by changing fermentation conditions such as pH and protein breakdown [37]. High tannin levels in forages were found to impair goat digestibility and feed intake in one study [38], but another study found that adding tannin in low concentrations to forages can improve goat weight by altering rumen fermentation and reducing internal parasite burdens as well as protecting dietary protein in the rumen [39]. Saponin and tannin can lower methane and ammonia production in the rumen, resulting in improved animal development and nutrition use [33].

### 4.2. Influence of CoCP on Digestibility and Intake 

Supplementation of CoCP at 100–200 g/d had no impact on feed intake except for an increase in EEI, which is a direct shift in the chemical composition of CoCP with around 3.5% EE content. Our findings reveal that plant-derived byproducts have only a minor impact on palatability. At all amounts of CoCP added to the diet, animals maintain a steady feed intake. Incorporation (not supplementation) of up to 200 g/d CoCP has no adverse effects on feeding value and may significantly increase the propionate:acetate ratio in the end products of fermentation and thereby increase amino acid uptake and reduce CH_4_ production. Similar to other studies that have shown that including coffee residues in diets at values between 25 and 28% had no negative impact on the DM digestibility of sheep [40], total nitrogen intake of dairy heifers [41], or feed intake of lambs during fattening [42].

However, supplementation with CoCP at 300 g/d can have a detrimental effect on digestion. This may be because the presence of tannin, saponin, and phenolic chemicals may influence the nutritional digestibility [14]. A reduction in rumen cellulolytic bacteria, in accordance with Patra et al. [33], may be what causes plant extracts’ inhibitory effects on feed digestibility. It could be related to the volatile essential oils, tannins, and other metabolites present in the plants under investigation that limit the action of certain enzymes (such as acetylesterase, xylanase, and carboxymethylcellulase) [43].

### 4.3. Influence of CoCP on Rumen Characteristics and Blood Metabolites

In this study, supplementation with CoCP decreased BUN in comparison with the control group (Linear, *p* < 0.01); although, likewise, in our experiment, the slight decrease in NH_3_-N was not statistically significant. These effects may be related to the high content of phenolic compounds, especially the amount of CT, SP, polyphenols, and caffeine [44]. 

The use of tannins in the ruminant diet is well recognized to inhibit protein breakdown in the rumen [45]. Furthermore, the data in Table 3 clearly demonstrated the positive effect of the maximum CoCP supplementation (300 g/d) on BUN release. NH_3_ synthesis in the rumen is closely associated with BUN concentrations [46]. Because blood urea N rises when the kidney is harmed, it is a useful indicator of renal function. 

The most commonly used biochemical measurements in evaluating energy metabolism in animals such as dairy cows and small ruminants are GLU, NEFA, and BHBA in plasma. According to one study, plasma NEFA and BHBA have been found to have a negative correlation to energy balance in lactating cows [47]. The increase in plasma glucose with increased CoCP intake revealed the glucogenic capacity of this additive in the current investigation [48], with a mean glucose amount of 65.79 mg/dL. These results agree with those of Chanjula et al. [39], who observed a mean value of 62.90 mg/dL in goats fed fermented oil palm fronds supplemented with urea-calcium hydroxide. Because the optimal glucose range is 50 to 80 mg/dL [46], all groups in all samples were within this range.

Furthermore, CoCP addition reduced the plasma content of NEFA and BHBA. These findings indicate CoCP could reduce body fat mobilization while also enhancing negative energy balance and milking ability in lactating cattle [49]. Lower NEFA concentrations in the CoCP treatment group might be attributed to higher content in the rumen. Hence, CoCP supplementation was expected to increase energy intake, which is consistent with the present research, and hence decrease the level of NEFA and BHBA in the serum. Improved negative energy balance related to CoCP serves as the foundation for its use in small ruminants. However, the mode of action is unknown. Creatinine is a biological marker used to evaluate muscle activity or renal disease. In the current research, it was discovered that supplementation of CoCP had no effect on creatinine levels during the administration of meals with or without CoCP, similar to the results reported by Carta et al. [50].

### 4.4. Influence of CoCP on Microbial Population

It is widely known that PSM has antibacterial activities against ruminal microbiota (bacteria, protozoa, and fungi). Both phenolic and nonphenolic chemicals are responsible for this activity [51]. The effect on the ruminal microbiota varies depending on the plant type ingested.

CoCP supplementation had no effect on ruminal microbial numbers, but it did appear to reduce protozoa populations linearly by about 20.2% as the amount of CoCP supplemented increased. Because of the hydrophobicity of their active chemicals, PSM has a wide spectrum of anti-microbial actions against microbial populations, including protozoa and PSM [44,45]. In their dairy cows’ study, Junior et al. [52] employed *A. mearnsii* containing CT at 6 g/kg, which led to a reduction in protozoal numbers. Nevertheless, research on the effect of tannins on ruminal protozoa has been contradictory [43]. While a few investigations found inconclusive results [8], others found significant antiprotozoal action [15,16]. The exact mechanism by which tannins influence protozoa in the rumen is unknown; however, tannins’ lipophilic nature has been proposed to increase EO permeability through the protozoal membrane [34]. In summary, PSM generally decreases the ruminal protozoal population because of a decrease in cell membrane permeability, resulting mostly in decreased ruminal CH_4_ generation (i.e., inhibition of ruminal methanogens). 

Furthermore, Patra and Saxena [33] demonstrated that SP-containing plants or extracts limit protozoal growth. Saponins could bind to cholesterol and form irreversible complexes in the protozoal cell membrane, resulting in cell membrane breakdown, lysis, and death [32]. Saponins have been shown to have antiprotozoal properties in vitro [32,33] and in vivo [37]. Furthermore, Fagundes et al. [53] discovered that tannin-rich fodder affected the population of rumen bacteria. Since bacteria are less preyed upon by protozoa, a drop in protozoal numbers increases the total bacterial population.

Because some methanogen communities retain ectosymbionts and endosymbionts of protozoa, the removal of protozoa by SP can diminish CH_4_ synthesis [32]. Furthermore, CT inhibits methanogenesis by either directly lowering methanogenic archaea or indirectly reducing protozoal numbers, hence lowering methanogens symbiotically connected with protozoal numbers [33,43].

On the other hand, CoCP supplementation at 300 g/head/day decreased the protozoal population. In this study, the relative impact of SP’s anti-protozoa impact in CoCP on acetate, propionate, and CH_4_ synthesis was not assessed. Since the model in this experiment estimated CH_4_, discussions on CH_4_ production with other components could be restricted. Although there are several models with various features for estimating CH_4_ emission from ruminants, the majority of them rely on feed intake, which is challenging to gather on a large scale, making them ineffective [27].

### 4.5. Influence of CoCP on Volatile Fatty Acid Profiles

Changes in ruminal fermentation end products such as NH_3_-N and VFA are generated by changes in ruminal bacteria caused by phytochemical supplementation [37,44]. The concentration of TVFAs in the rumen shows animal feeding efficiency, as TVFAs are the primary source of energy for ruminants. In this experiment, CoCP supplementation at 200 g/head/day improved propionate while decreasing acetate, C2:C3, and CH_4_ synthesis. Likewise, Gunun et al. [36] found that increasing propionate resulted in a decrease in CH_4_ generation by Mao seed meal (96 g CT/kg and 92 g SP/kg) at 0.8–2.4% DM intake in goats. Furthermore, Cherdthong et al. [51] reported that supplementing Thai native beef cattle with *Delonix regia* seed meal (93 g tannins/kg and 12 g SP/kg) at 90–270 g/head/day increased propionate synthesis while reducing CH_4_ generation at 4 h post-feeding. These observations are most likely because CT in MPM suppresses methanogen for more than 15 h, reducing CH_4_ generation and boosting propionate volume proportion in the rumen prior to feeding. CT in CoCP could reduce CH4 estimates by (a) suppressing the methanogen population directly, (b) changing the VFA profile, and (c) changing the C2:C3 ratios [51]. Declined ruminal total and cellulolytic bacterial numbers are frequently linked to reductions in VFA synthesis when tannins have a deleterious effect on substrate fermentation [45]. Tannins’ influence could be seen in changes to the primary VFA ratios [40]. With tannin feeding, there was an increase in acetate and a decrease in propionate or acetate [38].

Saponins have different effects on VFA production. The majority of investigations found that ruminal propionate levels were higher while acetate and butyrate levels were lower [33]. Such effects are caused by saponins’ inhibitory effects on G+ bacteria (typically acetate producers) and protozoa, leading to greater propionate production [54]. In an in vitro investigation, Hu et al. [55] demonstrated that tea saponins at 8 mg/0.2 g of diet had no effect on acetate, propionate, or butyrate levels.

### 4.6. Influence of CoCP on N Utilization

CoCP had no effect on N consumption, fecal N output, urine N excretion, total N excretion, or N absorption. Although retention of N was unaffected at 100 to 200 g/d, CoCP was reduced once 300 g/d of CoCP was added. The maximum nitrogen retention was obtained by supplementing with 100 to 200 g/d of CoCP (percent of N intake). Likewise, Wanapat et al. [46] discovered that although 150 g/d of lemongrass powder (LGP) had no effect on N consumption, the output of N, N absorption, or retention of N, 200 or 300 g/d of LGP addition reduced the intake of N, the output of N, absorption of N, and retention of N. Owens and Zinn [56] indicate that differences in N metabolism might be represented by N retention and N excretion since N retention was the most relevant marker of animals’ protein dietary status. Currently, 100 to 200 g/d of CoCP addition reduces N excretion while increasing N retention, with the N accumulation to N intake ratios of 9.31; 9.19 g/d and 56.14; 57.82%, respectively. This could be due to the addition of CoCP, which can limit the level of N breakdown in the rumen, resulting in slower N excretion rates when compared with the control [17,18]. Excessive N is promptly converted to NH_3_ by ruminal microbial urease, which is normally eliminated in the urine as urea [35,46].

## 5. Conclusions

According to this study, CoCP may be added to goat diets at a level of up to 200 g/d without compromising feed intake or nutrient digestibility. With 200 g/d of supplemental intake, TVFA and C3 are maximized. According to the study, 300 g/d is not recommended since it lowers digestibility and rumen fermentation output. At this level, N absorption also degrades. Future study into goat products, such as the impact of CoCP on carcasses or meat quality, is likely to lead to CoCP being utilized as a feed supplement in environmentally friendly animal production systems.

## Figures and Tables

**Figure 1 vetsci-09-00532-f001:**
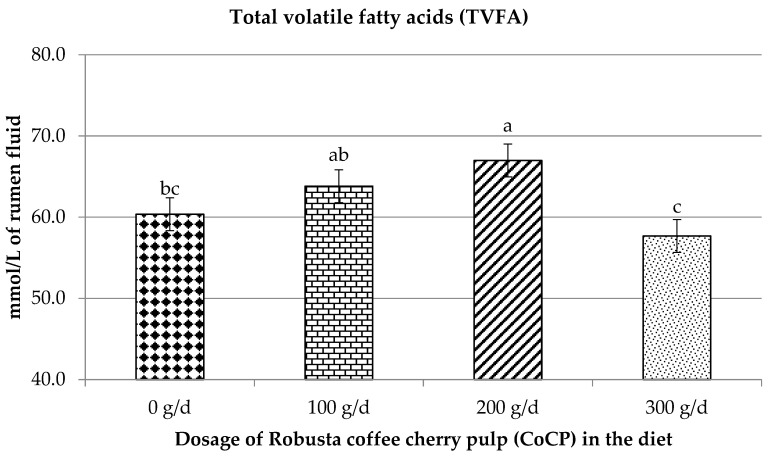
Effects of dietary CoCP supplementation (g/d) on total volatile fatty acids (TVFA) of Thai crossbred goats. ^a–c^ means in the same row with different lowercase letters differ (*p* < 0.05, *p* < 0.01).

**Figure 2 vetsci-09-00532-f002:**
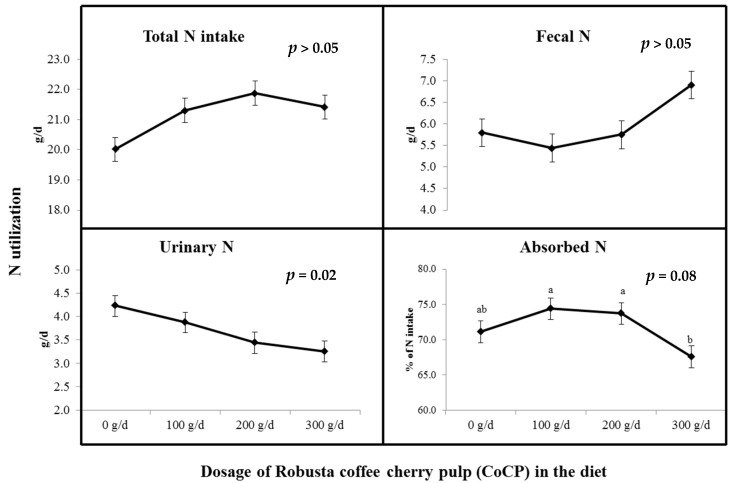
The figure plots the average outcomes of supplement levels (g/d) of CoCP on N balance in goats. ^a,b^ Different superscripts displayed significantly different mean values (*p* < 0.05).

**Table 1 vetsci-09-00532-t001:** Ingredients and chemical compositions used in the experiment: TMR basal diet and coffee cherry pulp (CoCP).

Item	Basal Diet	CoCP ^3^
Ingredients, %DM		
Rice straw	30.0	
Ground corn	45.0	
Soybean meal	7.30	
Fish meal	0.40	
Leucaena leaves meal	7.00	
Palm kernel cake	7.00	
Molasses	2.00	
Dicalcium phosphate	0.40	
Salt	0.20	
Mineral and vitamin mix ^1^	0.70	
Chemical composition, %		
DM	95.27	94.52
OM	94.60	99.03
CP	15.10	11.62
EE	1.90	3.52
NDF	47.33	43.58
ADF	24.34	10.41
GE, Mcal/kg DM	4.35	4.08
ME, Mcal/kg DM ^2^	2.75	2.34
Tannins, mg/100 g DM	-	4.27
Saponins, mg/100 g DM	-	2.61
Caffeine, mg/100 g DM		0.92
Antioxidant activity content (mg Fe (II) equivalent/ 100 g DM)	-	3.75
Phenolic compound content (mg gallic acid equivalent/ 100 g DM)	-	7.50
Flavonoid content (mg catechin equivalent/ 100 g DM)	-	4.37

DM, dry matter; CP, crude protein; OM, Organic matter; EE, Ether extract; NDF, Neutral detergent fiber; ADF, Acid detergent fiber; ME, Metabolizable energy; GE, Gross energy. ^1^ Minerals and vitamins (each kg contains): Vitamin A: 10,000,000 IU; Vitamin E: 70,000 IU; Vitamin D: 1,600,000 IU; Fe: 50 g; Zn: 40 g; Mn: 40 g; Co: 0.1 g; Cu: 10 g; Se: 0.1 g; I: 0.5 g. ^2^ Calculated value. ^3^ CoCP, Robusta coffee cherry pulp.

**Table 2 vetsci-09-00532-t002:** Effects of dietary CoCP supplementation (g/d) on feed intake, nutrient intake, and apparent digestibility of nutrients in goats.

Item	Supplement Levels (g/d) of CoCP ^1^				SEM ^2^	Contrast *p*-Value ^3^	
	0	100	200	300		Linear	Quadratic
Dry matter intake							
Total DMI, kg/d	0.851	0.885	0.898	0.869	0.04	0.75	0.50
DMI, %BW	3.51	3.84	3.87	3.58	0.16	0.71	0.04
DMI, g/kg W^0.75^	77.84	84.14	84.90	79.32	3.50	0.69	0.06
Nutrient intake, kg/d							
OMI	0.786	0.819	0.824	0.786	0.03	0.97	0.41
CPI	0.126	0.133	0.137	0.132	0.01	0.74	0.46
EEI	0.016 ^b^	0.018 ^ab^	0.022 ^a^	0.022 ^a^	0.01	<0.01	0.41
NDFI, kg/d	0.351	0.376	0.390	0.410	0.03	0.07	0.93
ADFI, kg/d	0.186 ^b^	0.201 ^ab^	0.225 ^a^	0.244 ^a^	0.01	0.04	0.91
Apparent digestibility, %							
DM	69.38 ^a^	71.10 ^a^	70.21 ^a^	63.82 ^b^	1.15	0.03	0.03
OM	70.95 ^a^	72.28 ^a^	70.59 ^a^	64.73 ^b^	1.27	0.01	0.04
CP	71.35 ^ab^	74.42 ^a^	73.72 ^a^	67.16 ^b^	1.30	0.11	0.02
EE	70.10 ^b^	79.61 ^a^	83.40 ^a^	81.29 ^a^	2.56	<0.01	0.03
NDF	56.93 ^a^	59.07 ^a^	57.56 ^a^	50.44 ^b^	1.20	0.09	0.05
ADF	34.65 ^a^	38.94 ^a^	37.41 ^a^	25.51 ^b^	2.50	0.06	0.04
Estimated energy intake ^4^							
ME, Mcal/d	2.12	2.25	2.21	1.94	0.12	0.30	0.12
ME, Mcal/kg DM	2.49 ^a^	2.54 ^a^	2.46 ^a^	2.23 ^b^	0.04	<0.01	0.02

DMI, dry matter intake; CPI, crude protein intake; OMI, Organic matter intake; EEI, Ether extract intake; NDFI, Neutral detergent fiber intake; ADFI, Acid detergent fiber intake; DM, dry matter; CP, crude protein; OM, Organic matter; EE, Ether extract; NDF, Neutral detergent fiber; ADF, Acid detergent fiber; ME, Metabolizable energy. ME, Metabolizable energy; GE, Gross energy. ^1^ Levels of coffee cherry pulp (CoCP) supplementation at 0, 100, 200, and 300 g/d/animal. ^2^ SEM = Standard error of the mean. ^3^ Significance was defined as *p* < 0.05, whereas *p* < 0.10 indicated a trend. ^4^ 1 kg of digestible organic matter (DOM) = 3.8 Mcal ME/kg [25]. ^a,b^ means in the same row with different lowercase letters differ (*p* < 0.05, *p* < 0.01).

**Table 3 vetsci-09-00532-t003:** Effects of dietary CoCP supplementation (g/d) on the rumen fermentation and blood metabolites in goats.

Item	Supplement Levels (g/d) of CoCP ^1^				SEM ^2^	Contrast *p*-Value ^3^	
	0	100	200	300		Linear	Quadratic
Temperature, °C	39.30	39.40	39.30	39.10	0.16	0.83	0.25
Ruminal pH	6.00	6.04	6.04	5.96	0.05	0.76	0.48
NH_3_-N, mg/dL	20.58	23.47	22.36	18.26	2.77	0.66	0.40
BUN, mg/dL	21.58 ^a^	17.39 ^b^	17.87 ^b^	17.37 ^b^	0.34	<0.01	<0.01
GLU, mg/dL	63.69 ^b^	66.25 ^ab^	66.75 ^a^	66.50 ^ab^	0.81	0.04	0.43
PCV, %	28.63	29.25	29.50	29.37	0.69	0.43	0.59
NEFA, µmol/L	225.63 ^a^	210.13 ^ab^	205.63 ^b^	198.88 ^b^	3.25	<0.01	0.67
BHBA, µmol/L	0.48 ^a^	0.42 ^ab^	0.35 ^b^	0.38 ^ab^	0.02	0.05	0.24
Cr, mg/dL	1.19	1.29	1.14	1.17	0.06	0.70	0.78

NH_3_-N, ammonia nitrogen; BUN, blood urea nitrogen; GLU, blood glucose; PCV, pack cell volume; NEFA, non-esterified fatty acids; BHBA, beta-hydroxybutyric acid; Cr, creatinine. ^1^ Level of coffee cherry pulp (CoCP) supplementation at 0, 100, 200, and 300 g/d/animal. ^2^ SEM = Standard error of the mean. ^3^ Significance was defined as *p* < 0.05, whereas *p* < 0.10 indicated a trend. ^a,b^ means in the same row with different lowercase letters differ (*p* < 0.05, *p* < 0.01).

**Table 4 vetsci-09-00532-t004:** Effects of dietary CoCP supplementation (g/d) on the ruminal volatile fatty acid (VFA) profiles and methane (CH_4_) estimation in goats.

Item	Supplement Levels (g/d) of CoCP ^1^				SEM ^2^	Contrast *p*-Value ^3^	
	0	100	200	300		Linear	Quadratic
VFA profiles, mol/100 mol							
Acetic acid	62.28 ^a^	58.40 ^b^	56.99 ^b^	57.35 ^b^	0.74	0.03	0.18
Propionic acid	22.58 ^c^	26.38 ^b^	28.04 ^a^	26.35 ^b^	0.36	0.02	0.03
Butyric acid	12.76	12.91	12.24	13.39	1.33	0.79	0.63
Acetic/propionic acid ratio	2.78 ^a^	2.28 ^b^	2.09 ^c^	2.23 ^bc^	0.04	0.01	0.04
Estimated methane, mol/100 mol ^4^	26.92 ^a^	24.18 ^b^	22.83 ^c^	23.91 ^b^	0.29	0.01	0.04

^1^ Levels of coffee cherry pulp (CoCP) supplementation at 0, 100, 200, and 300 g/d/animal. ^2^ SEM = Standard error of the mean. ^3^ Significance was defined as *p* < 0.05, whereas *p* < 0.10 indicated a trend. ^4^ Methane = (0.45 × acetic acid, mmol/L) – (0.275 × propionic acid, mmol/L) + (0.40 × butyric acid, mmol/L) [27]. ^a–c^ means in the same row with different lowercase letters differ (*p* < 0.05, *p* < 0.01).

**Table 5 vetsci-09-00532-t005:** Effects of dietary CoCP supplementation(g/d) on the microorganism count in the rumen.

Item	Supplement Levels (g/d) of CoCP ^1^				SEM ^2^	Contrast *p*-Value ^3^	
	0	100	200	300		Linear	Quadratic
Bacteria (×10^10^ cells/mL)	7.27	7.39	7.63	7.77	0.36	0.33	0.98
Fungal zoospores (×10^6^ cells/ mL)	1.10	1.05	1.28	1.26	0.12	0.23	0.91
Total protozoa (×10^6^ cells/mL)	3.02	2.99	2.81	2.41	0.26	0.06	0.95

^1^ Levels of coffee cherry pulp (CoCP) supplementation at 0, 100, 200, and 300 g/d/animal. ^2^ SEM = Standard error of the mean. ^3^ Significance was defined as *p* < 0.05, whereas *p* < 0.10 indicated a trend.

## Data Availability

Not applicable.

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
