# Peer review of "Digestibility, Blood Parameters, Rumen Fermentation, Hematology, and Nitrogen Balance of Goats after Receiving Supplemental Coffee Cherry Pulp as a Source of Phytochemical Nutrients"

_vetsci, 2022, doi:10.3390/vetsci9100532_

Round 1

Reviewer 1 Report

The authors demonstrated the effect of coffee cherry pulp (CoCP) to improve the animal performance mostly based on modulating rumen fermentation in goats.

This study is quite straightforward to test the effect of one of the important by-products in tropical area which contain a variety of phytochemicals.

I have added my comments below.

Please add the levels of CoCP supplementation in the abstract.

Line 34 - 36: Please designate the direction of linear effects on each VFA component and ratio

Line 68: Please add adequate references to support the effect of PTN, particularly of tannin ans saponin on protozoa and methane production in the rumen.

Line 68 - 70: Please add adequate reference to support the role of rumen protozoa.

Line 70: what does it indicate? Please rewrite the sentence to clarify which feed source contain those nutrient composition.

Line 113: Please add references which were used to select doses of CoCP for this study.

Line 116: What was inoculated? Please provide details.

Line 116: What is AD3E? Please provide information on this vitamin source.

Line 130: ad libitum -> ad libitum. Check the words which need to be italicized.

Line 181: What is previously prepared solution?

There is no description of how the authors measured zoospore.

Line 211: Specify the P-value range for trend

Line 393: On other hand -> On the other hand

Line 395: Though CH4 is not actually measured but calculated in this experiment, the authors should declare the limitation of their discussions on methane production.

Table 5

cell -> cells/ml

Total protozoa should be counted as cells/ml

Figure 2

Better to include all the P-values corresponded to each measurement in the figure 

Author Response

Response to Reviewer 1

Please add the levels of CoCP supplementation in the abstract.

Response:  Thank you for your suggestion, I have added already. Please see in manuscript.

Line 34 - 36: Please designate the direction of linear effects on each VFA component and ratio

Response:  Thank you so much, we have added. Please see in manuscript.

Line 68: Please add adequate references to support the effect of PTN, particularly of tannin ans saponin on protozoa and methane production in the rumen.

Response:  The references have been added. Please see in manuscript.

Line 68 - 70: Please add adequate reference to support the role of rumen protozoa.

Response:  The reference has been added. Please see in manuscript.

Line 70: what does it indicate? Please rewrite the sentence to clarify which feed source contain those nutrient composition.

Response:  We have modified as “Additionally, COCP contains significant amounts of neutral detergent fiber (NDF, 41.5–46.0%), acid detergent fiber (ADF, 37.8–41.0%), crude protein (CP, 9.7–10.3%), 1.0–2.0% of lipids, and 45.0–89.0% of carbohydrates (including pectin), suggesting the possibility of using it as ruminant feed [8–10].” Please see in manuscript Line 71-74.

Line 113: Please add references which were used to select doses of CoCP for this study.

Response:   We have provided as “Based on a coffee literature review on by-product utilization, daily CoCP supplementation at 0, 100, 200, and 300 g/day was used as the dietary treatment [12,15,16,17].”

Line 116: What was inoculated? Please provide details.

Response:  Thank you so much, in was changed to “injected”

Line 116: What is AD3E? Please provide information on this vitamin source.

Response:  Thank you so much. It is vitamin and we have defined. Please see in manuscript.

Line 130: ad libitum -> ad libitum. Check the words which need to be italicized.

Response:  Thank you so much, we have done. Please see in manuscript.

Line 181: What is previously prepared solution?

Response:  Thank you for your suggestion. We have modified as “Collect about 20 mL of ruminal fluid with a large-bore pipette (to allow for bacteria, protozoa, and zoospore) and add 1 mL to the formalin solution.”  Please see in manuscript Line 184-186.

There is no description of how the authors measured zoospore.

Response:   Thank you. It was modified to “Moreover, the total direct count of fungal were made using the methods of Galyean [28] based on the use of a haemacytometer (Booeco).” Please see in manuscript Line 195-196.

Line 211: Specify the P-value range for trend

Response: Thank you so much. We have added. Please see in manuscript Line 71-74.

Line 393: On other hand -> On the other hand

Response:   Thank you so much, we have revised. Please see in manuscript Line 71-74.

Line 395: Though CH4 is not actually measured but calculated in this experiment, the authors should declare the limitation of their discussions on methane production.

Response:   Thank you. We have discussed as “In this study, the relative impact of SP's anti-protozoa impact in CoCP on acetate, propionate, and CH4 synthesis was not assessed. Since the model in this experiment estimated CH4, discussions on CH4 production with other components could be restricted. Although there are several models with various features for estimating CH4 emission from ruminants, the majority of them rely on feed intake, which is challenging to gather on a large scale, making them ineffective [27].” Please see in manuscript.

Table 5 cell -> cells/ml

Response:    Thank you so much, we have modified.

Table 5 Total protozoa should be counted as cells/ml,

Response:    Thank you for your suggestion, we have revised. Please see in manuscript

Figure 2 Better to include all the P-values corresponded to each measurement in the figure

Response:    Thank you so much, we have revised. Please see in manuscript.

We thank you for your effort and provide valuable comments and suggestion.

Reviewer 2 Report

1. pls Keep the same font size used for author names

2. The title could be further improved. Blood parameters, ruminal fermentation, 

3. The abstract needs to be rewritten. There are a lot confusing and confilicting descriptions . "Nutrient or dry matter intake (DMI) was unaffected by CoCP supplementation (organic matter and crude protein)." what does this sentence mean ?

 4."there was a linear decrease in protozoa"  what is the unit of protozoa? count?5.the chemical composition of the specific TMR must be provided. 

Author Response

Response to Reviewer 2

  1. pls Keep the same font size used for author names

Response:    Thank you so much, we have revised. Please see in manuscript.

  1. The title could be further improved. Blood parameters, ruminal fermentation,

Response:    Thank you for your suggestion, we have revised as “Digestibility, blood parameters, rumen fermentation, hematology, and nitrogen balance of goats after receiving supple-mental coffee cherry pulp as a source of phytochemical nutrients”. Please see in manuscript

  1. The abstract needs to be rewritten. There are a lot confusing and confilicting descriptions . "Nutrient or dry matter intake (DMI) was unaffected by CoCP supplementation (organic matter and crude protein)." what does this sentence mean ?

Response:    We have modified to “Dry matter intake (DMI), organic matter intake (OMI), and crude protein intake (CPI) were unaffected by the addition of CoCP.” Please see in manuscript.

4."there was a linear decrease in protozoa"  what is the unit of protozoa? count?

Response: Thank you so much, it was protozoa count. We have modified sentence as “With increased CoCP supplementation, there was a linear decrease in protozoa count by about 20.2 % as the amount of CoCP supplemented increased (P = 0.06).” Please see in manuscript.

5.the chemical composition of the specific TMR must be provided.

Response: Thank you so much, and the chemical composition of TMR has been indicated in Table 1 already. Please see.

We thank you for your effort and provide valuable comments and suggestions.

Reviewer 3 Report

This paper describes a conventional feeding trial. The design and analysis are generally satisfactory and the results are worthy of publication. However, the discussion is far too long and many of the references are not strictly relevant to this study. (It reads as though it has been cut and pasted from a PhD thesis).  For example, the section from l 318-329 including phrases like ‘use of feed additives on a regular basis generally results in an adequate response’   is largely meaningless. The discussion should focus on the important conclusions, namely that incorporation (not supplementation ) of up to 200g/d CoCP has no adverse effects on feeding value and may significantly increase the propionate:acetate ratio in the end products of fermentation and thereby increase amino acid uptake and reduce methane production. These observations are worthy of further study.

Specific comments

You describe the incorporation of CoCP as a food supplement. As I understand it CoCP was fed together with the basal diet (the two being very similar in proximal constituents and estimated ME) up to an incorporation level of about 30%. Is this correct? If so, it cannot be described as a supplement.

Replace the word ‘enhanced’ by ‘affected’ whenever it appears. Some of the claimed effects cannot be described as enhancement.

Table 1. Superscripts in the notes to the table are not explained

Fig 1. I don’t understand this. The y axis is defined as total VFA and expressed as mmol/l. Per litre of what?

Author Response

Response to Reviewer 3

This paper describes a conventional feeding trial. The design and analysis are generally satisfactory and the results are worthy of publication. However, the discussion is far too long and many of the references are not strictly relevant to this study. (It reads as though it has been cut and pasted from a PhD thesis).  For example, the section from l 318-329 including phrases like ‘use of feed additives on a regular basis generally results in an adequate response’   is largely meaningless. The discussion should focus on the important conclusions, namely that incorporation (not supplementation ) of up to 200g/d CoCP has no adverse effects on feeding value and may significantly increase the propionate:acetate ratio in the end products of fermentation and thereby increase amino acid uptake and reduce methane production. These observations are worthy of further study.

Response:    Thank you. We have tried our best to modified regarding to your comments. Please see in manuscript.

You describe the incorporation of CoCP as a food supplement. As I understand it CoCP was fed together with the basal diet (the two being very similar in proximal constituents and estimated ME) up to an incorporation level of about 30%. Is this correct? If so, it cannot be described as a supplement.

Response:    CoCP was top-up supplemented daily at 0, 100, 200, and 300 g/day and TMR fed ad libitum. To verify that the animals received all the supplements, CoCP was manually hand-mixed with 200 g of TMR feed before the morning meal, and more TMR was pro-vided to individual animals later.

Replace the word ‘enhanced’ by ‘affected’ whenever it appears. Some of the claimed effects cannot be described as enhancement.

Response:    Thank you for your suggestion, we have modified.

Table 1. Superscripts in the notes to the table are not explained

Response:  The superscripts of 1-3 have been defined in footnote already.

Fig 1. I don’t understand this. The y axis is defined as total VFA and expressed as mmol/l. Per litre of what?

Response:  It stands for mmol/L of rumen fluid. We have added a unit in Figure 1.

We thank you for your effort and provide valuable comments and suggestion.

Reviewer 4 Report

Review comments on “Digestibility, rumen fermentation, hematology, and nitrogen balance of goats after receiving supplemental coffee cherry pulp as a source of phytochemical nutrients” by Felista W. Mwangi et al.

This paper presents study the impact of adding dried coffee cherry pulp (CoCP) to goat feed on the digestibility of the feed, rumen fermentation, hematological, and nitrogen balance.

My main general comments are as below:

- Written English requires further editing. The manuscript suffers from typing and grammar errors and should be thoroughly read by an expert or English native speaker to ensure that your paper's text is polished and easy to read.

- This manuscript cited too many not highly relevant references to the reported study, but on the other side not enough highly relevant references on the particular study the author(s) presented. Please correct this issue by removing less relevant references but adding a few (more) highly agricultural applications relevant references, to keep the total number of references around 50.

- There should be correct formatting of formulas (located in the center) and there should be a correct description of the symbols in the formulas. Authors should refer to the examples and requirements of the journal. For example, the formula on 206 line should have a reference number. On line 207, the Yijk should have a lower index ijk. Authors should check and revise all text.

- On line 179, there is no closing parenthesis “)”. Authors should check and revise all text.

- On line 32, “DMI (% BW; P = 0.04)”, on line 218 “DMI % BW (P = 0.04)”, and “DMI BW^0.75 (P = 0.06)”. Authors should use a single understandable notation style.

Author Response

Response to Reviewer 4

- Written English requires further editing. The manuscript suffers from typing and grammar errors and should be thoroughly read by an expert or English native speaker to ensure that your paper's text is polished and easy to read.

Response:  Thank you for the recommendation. In the present revised version, we have asked the professionals in the field who are experts in English grammar. Please see the modification in the manuscript.

- This manuscript cited too many not highly relevant references to the reported study, but on the other side not enough highly relevant references on the particular study the author(s) presented. Please correct this issue by removing less relevant references but adding a few (more) highly agricultural applications relevant references, to keep the total number of references around 50.

Response:  Thank you for your suggestion. References have been reduced from 91 citations to 56 references.

- There should be correct formatting of formulas (located in the center) and there should be a correct description of the symbols in the formulas. Authors should refer to the examples and requirements of the journal. For example, the formula on 206 line should have a reference number. On line 207, the Yijk should have a lower index ijk. Authors should check and revise all text.

Response:  Thank you we have modified and please see the formula used.

- On line 179, there is no closing parenthesis “)”. Authors should check and revise all text.

Response:  Thank you for your suggestion, we have revised it.

- On line 32, “DMI (% BW; P = 0.04)”, on line 218 “DMI % BW (P = 0.04)”, and “DMI BW^0.75 (P = 0.06)”. Authors should use a single understandable notation style.

Response:  Thank you for your suggestion, we have revised it.

We thank you for your effort and provide valuable comments and suggestions.

Round 2

Reviewer 1 Report

Thanks for accepting my comments.

Reviewer 2 Report

pls check comments from other reviewers

Reviewer 4 Report

All comments and errors have been corrected